# An FPGA-Oriented Baseband Modulator Architecture for 4G/5G Communication Scenarios

**Mário Lopes Ferreira [1,2,*]** and **João Canas Ferreira [1,2]**

[1] Institute for Systems and Computer Engineering, Technology and Science (INESC TEC),
Rua Dr. Roberto Frias, s/n 4200-465 Porto, Portugal; jcf@fe.up.pt

[2] Faculty of Engineering of the University of Porto, Rua Dr. Roberto Frias, s/n 4200-465 Porto, Portugal

[*] Correspondence: mariolf@fe.up.pt

**Abstract:** The next evolution in cellular communications will not only improve upon the performance of previous generations, but also represent an unparalleled expansion in the number of services and use cases. One of the foundations for this evolution is the design of highly flexible, versatile, and resource-/power-efficient hardware components. This paper proposes and evaluates an FPGA-oriented baseband processing architecture suitable for communication scenarios such as non-contiguous carrier aggregation, centralized Cloud Radio Access Network (C-RAN) processing, and 4G/5G waveform coexistence. Our system is upgradeable, resource-efficient, cost-effective, and provides support for three 5G waveform candidates. Exploring Dynamic Partial Reconfiguration (DPR), the proposed architecture expands the design space exploration beyond the available hardware resources on the Zynq xc7z020 through hardware virtualization. Additionally, Dynamic Frequency Scaling (DFS) allows for run-time adjustment of processing throughput and reduces power consumption up to 88%. The resource overhead for DPR and DFS is residual, and the reconfiguration latency is two orders of magnitude below the control plane latency requirements proposed for 5G communications.

**Keywords:** reconfigurable hardware; FPGA; dynamic partial reconfiguration; baseband processing; carrier aggregation; 4G/5G coexistence; cloud-RAN

## 1. Introduction

The 5G New Radio (NR) horizon is getting closer, and its first Physical layer (PHY) specification (3GPP Release 15) defines support for services and use cases that can be classified into three categories: enhanced Mobile Broadband (eMBB), Ultra-Reliable Low Latency Communications (URLLC), and massive Machine-Type Communications (mMTC) [1]. While eMBB (e.g., Gigabit per second peak rates, 3D video, and UHD screens) is characterized by high data rates and increased capacity, URLLC services (e.g., self-driving cars and tactile Internet) are latency-sensitive and highly reliable; whereas mMTC (e.g., smart cities and Internet-of-Things) aim at high density and sporadic short packet transmission. Thus, waveform numerologies for 5G NR must be flexible and scalable not only to support this diversified range of services and requirements, but also to allow the exploitation of spectrum bands previously unused, such as millimeter-wave (mmWave) bands [2].

Although 3GPP Release 15 uses Cyclic Prefix OFDM (CP-OFDM) as the basis for the 5G NR physical layer design, several new OFDM-based waveforms have been discussed and proposed for use alongside CP-OFDM for specific use cases [1]. Filter-Bank Multi-Carrier modulation (FBMC), Generalized Frequency Division Multiplexing (GFDM), and Universal Filtered Multi-Carrier modulation (UFMC) are among the strongest waveform candidates to be exploited in future 5G use cases. Multi-waveform coexistence is indeed a likely near-future scenario with a "few 5G candidate

waveforms (FBMC, UFMC, GFDM) intermingled with a variety of 3G and 4G waveforms" [3]. In heavily used portions of the spectrum, such as the sub-6 GHz band, multi-waveform coexistence requires Dynamic Spectrum Access (DSA) and Carrier Aggregation (CA) techniques to achieve a more efficient spectrum utilization [1].

Other important factors in 5G NR are cost minimization and energy efficiency. Cloud Radio Access Networks (C-RANs) attempt to achieve a more efficient energy consumption and resource allocation by deploying a central Baseband processing Unit (BBU) that serves multiple Remote Radio Heads (RRHs) [4]. This network architecture relies on reconfigurable hardware modules to implement BBUs supporting the different access technologies and modes of operation used by RRHs.

The realization of the 5G vision will strongly rely on the design of hardware infrastructure adjusted to the challenges imposed by future wireless communications. Regarding digital baseband processing, hardware designs should be: (1) flexible to support multiple services requirements, waveforms, and numerologies; (2) evolvable/forward-compatible to be easily upgradeable with new functionalities and future modes of operation, thus extending the system's duty lifetime; (3) energy-efficient and (4) cost-effective. FPGAs are convenient platforms to design systems with these characteristics. Apart from their capacity for parallel intensive computation, FPGAs feature a high degree of flexibility not only at design time, but also at runtime, through Dynamic Partial Reconfiguration (DPR) [5]: the ability to reconfigure portions of FPGA logic fabric (reconfigurable regions) while the other portions remain unchanged and running. This increases the FPGA functional density, as mutually-exclusive circuits/functionalities are implemented on the same hardware resources at different instants [6]. Ultimately, this allows for the use of a smaller device to implement a featureful application, thus enabling cost and power savings. Additionally, the functionality of a DPR-based design can be extended by simply designing new configurations for the reconfigurable regions, without the need to redesign the static region.

This work proposes an FPGA-oriented architecture for baseband downlink transmission suitable for DSA, C-RAN, and 4G/5G coexistence scenarios. The architecture supports three 5G waveform candidates (OFDM, FBMC, and UFMC). From these waveforms, UFMC is the one with fewer hardware implementations proposed and discussed in the literature. The baseband modulators for the selected 5G waveforms are implemented on FPGA reconfigurable regions. The functionality of these regions can be customized at run-time through DPR. Although only two modes of operation are considered for each waveform, new modes or even waveforms can be added in the future by generating partial bitstreams for the corresponding reconfigurable regions. In our system, three independent and reconfigurable baseband processors can be simultaneously used to process different component carriers in CA, multiple waveforms in coexistence scenarios, or different access modes in C-RANs. Besides DPR, the proposed architecture employs Dynamic Frequency Scaling (DFS) [7] in order to adapt throughput and power consumption to the instant communication requirements. The results highlight the resource efficiency achieved with DPR at the cost of a tolerable reconfiguration latency that is below 1 ms per baseband processor; and the power efficiency of DFS with negligible latency and resource overhead.

This paper's contributions are: (1) a flexible, reconfigurable, and adaptable FPGA-oriented architecture for baseband transmission in 5G communication scenarios combining DPR with DFS; (2) a qualitative and quantitative evaluation of the impact of DPR and DFS in this type of application regarding adaptation latency and resource overhead; and (3) a low-latency UFMC modulator scheme with multiplier-less FIR filtering to reduce embedded DSP block utilization.

After discussing baseband processing in 5G scenarios and reviewing reconfigurable baseband processor implementation in Section 2, the proposed baseband modulator architecture is described (Section 3) and evaluated (Section 4). The obtained results are discussed and conclusions are finally presented in Section 5.

## 2. Background and Related Work

This section discusses the impact of some aspects of digital baseband processing in 5G scenarios and reviews related work on reconfigurable FPGA-oriented baseband processors for multi-waveform and flexible wireless communications.

Carrier aggregation (CA) is a technique with the potential to increase system capacity and enhance spectrum usage. It is already exploited in 4G/LTE communications, and 5G will surely rely on it to improve spectrum efficiency within heavily used frequency bands and also across different frequency bands (from low-band to mmWaves) [8]. There are two main types of CA: contiguous CA and non-contiguous CA. In the former case, aggregated component carriers are adjacent to each other, whereas in the latter case, they are fragmented along the spectrum. Although contiguous CA does not require deep changes in the transceiver PHY, multiple and flexible PHY blocks are needed for non-contiguous CA, in order to adaptively tune communication parameters for each component carrier to aggregate [9].

Another technique that can be combined with CA to boost spectrum utilization efficiency is Dynamic Spectrum Access (DSA): a mechanism through which idle spectrum resources (spectrum holes) from different primary (licensed) users are opportunistically accessed by secondary (non-licensed) users [10]. To avoid interferences with primary users, secondary users monitor and detect available spectrum holes through spectrum sensing techniques.

Due to the spectral agility and multitude of 5G service requirements, it is unfeasible and inefficient to design separate standalone radio systems for each service or use case. Instead, a 5G baseband system should be flexible enough to support and multiplex a wide variety of services. This is also the case for C-RAN architectures, where the centralized baseband processing should be carried out by adaptive and reconfigurable hardware according to the concept of software-defined radio [11].

Prior to the advent of 5G communications, the growing interest in SDR and cognitive radio led to research efforts towards flexible and reconfigurable baseband processors. Apart from the trade-offs between GPU flexibility and ASIC performance, the possibility for DPR in SRAM-based FPGAs makes them compelling hardware platforms for baseband architectures for agile and flexible radio communications [11–13]. Early works exploited DPR for the design of specific modules in the baseband processing chain, like FIR filters or constellation mappers [14,15]. One of the first multi-waveform flexible PHY architectures for SDR transmitters was proposed by He et al. [16]. This architecture was implemented on a Xilinx Virtex-5 FPGA device, and it combines two reconfiguration techniques: (a) DPR is employed to dynamically change the baseband processing mode of operation; and (b) DFS is used to adapt the clock frequency of the digital up-converter and the baseband processor. The implementation supports two waveforms (OFDM and WCDMA ) and several 3G/4G standards and modes of operation. This work focuses on the resource efficiency achieved with DPR compared with an equivalent static multi-mode design, but lacks a comprehensive analysis of the DPR/DFS latency or the impact of DFS on power consumption. Pham et al. [17] also exploited DPR in a multi-standard OFDM transceiver implementation. These works paid little attention to the impact of CA on the PHY, as the architectures considered a single reconfigurable baseband processor only.

As discussion about 5G progressed, the interest in alternative waveforms to mitigate OFDM weaknesses increased [1], and multi-waveform coexistence scenarios are likely to be a reality in future communications [3]. For instance, Kaltenberger et al. [18] presented a scenario combining DSA with multi-waveform coexistence, where 5G networks use the existing 4G-LTE infrastructure as an anchor. While a primary 4G/LTE system operates, secondary 5G systems exploit spectrum holes via DSA and use different waveforms (e.g., OFDM, GFDM, and UFMC) for transmission without affecting primary user communications.

The first flexible hardware platform designed for multiple 5G waveform scenarios was proposed in [19]. It is a complete transmitter/receiver platform that comprises hardware and software modules for digital baseband processing, RF boards, and high-level software applications for system control and information display purposes. Baseband processing on the transmitter side is implemented on

a Xilinx Zynq xc7z020 device that includes an FPGA and an embedded ARM processor on the same chip. Three 5G waveform candidates are supported—OFDM, FBMC, and UFMC—and a baseband modulator for each waveform is implemented. The high-level software application selects the type of waveform and the baseband parameters to be used. Then, a DMA controller fetches data stored on the external DDR memory and sends it to the selected waveform modulator. In turn, the modulator performs baseband processing and forwards the results to an RF front-end extension board.

At the baseband processing level, the flexibility in [19] comes from multiplexing between the three baseband modulators: the static multi-mode approach. Thus, although the three modulators are always present in the system, only one can be used at the same time. This is not efficient from a resource utilization perspective and not suitable for non-contiguous CA scenarios. For each waveform, two numerologies are supported by defining the value of some baseband parameters. Therefore, the modulators have to be designed for the most demanding scenario. This static multi-mode approach is also not upgradeable. If new waveforms or numerologies have to be supported in the future, the FPGA baseband infrastructure has to be redesigned. Moreover, this approach is not scalable, as the continuing addition of new modes of operation may force design migration to a larger device, with an associated overhead in terms of cost and power consumption.

OFDM popularity over the last two generations of cellular communications led to numerous works on FPGA-based implementations for OFDM baseband modulators (e.g., [20,21]. We have also contributed with a dynamically-reconfigurable OFDM modulator for LTE downlink transmission employing DPR [22] and a parallel-pipelined architecture OFDM modulator that exploits DFS to support scalable numerologies for 5G communications [23].

Regarding FBMC and UFMC, there are few published works on hardware implementations for baseband modulators. Nadal et al. [24] and Berg et al. [25] presented FPGA-based implementations for polyphase network FBMC modulators. Alternatively, we showed that, despite its higher computational complexity, the Frequency Spreading (FS) approach is a convenient scheme for FPGA-based FBMC designs and proposed a flexible and resource-efficient variant of an FS-FBMC baseband modulator [26].

For UFMC, Medjkouh, et al. [27] presented an FPGA-based implementation for a baseband modulator that exploits the separation between sub-band and subcarrier processing and the decomposition of the UFMC symbol into prefix, core, and suffix parts. With this technique, the authors claim a significant complexity reduction compared to a baseline UFMC implementation, which can be very important when the number of allocated sub-bands for UFMC transmission increases. That work followed the numerology for LTE 5 MHz channelization and presented resource utilization results for post-synthesis (before place-and-route) design only. In turn, Jafri et al. [28] followed the algorithm proposed in [29] and presented an FPGA-based UFMC modulator for LTE 10 MHz channelization. Each UFMC sub-band is not processed in parallel, but using a ping-pong buffering/memory strategy, which requires clock frequencies above 300 MHz to achieve acceptable processing latencies.

Our work presents an architecture composed of tiled baseband processor blocks that can be turned on/off and whose operation can be customized at run-time according to communication demands. While previously-published works focused on one or a few communication features and scenarios, our design combines enhanced flexibility, upgradeability, cost-effectiveness, and energy-efficiency into a versatile baseband processing architecture suitable for downlink transmission in 5G scenarios such as: dynamic carrier aggregation, centralized multi-mode baseband processing, multi-waveform, and multi-service coexistence. Regarding UFMC baseband modulation, our approach considers a sporadic short-packet transmission scenario using few filtered sub-bands and aims to reduce the utilization of FPGA-embedded DSP blocks in order to fit the design in cost-optimized FPGA devices.

## 3. Baseband Modulator Architecture

This section presents our proposed baseband modulator architecture. First, the modulators for each selected 5G waveform are described; then, the top-level reconfigurable architecture is presented. Our baseband modulator was implemented on a cost-optimized Zynq xc7z020 device.

### 3.1. Baseband Modulation for 5G Waveform Candidates

The waveforms supported by the baseband modulator are OFDM, FBMC, and UFMC. These multi-carrier waveforms efficiently perform waveform synthesis using the Inverse Fast Fourier Transform (IFFT) operation. The differences between the selected waveforms are mainly related to the techniques adopted for time-domain windowing (pulse-shaping in the frequency domain) and/or time-domain filtering (equivalent to frequency domain windowing).

OFDM is the most prominent waveform in current wireless communications, and it is characterized by the orthogonality between subcarriers, which eliminates inter-carrier interference. Every OFDM symbol is prepended with a Cyclic Prefix (CP), which mitigates inter-symbol interference, but contributes to the degradation of spectral efficiency. Currently, 4G LTE systems improve the frequency response of Cyclic Prefix OFDM (CP-OFDM) by applying time-domain windowing of the CP-extended OFDM symbols and overlapping the edge transition of adjacent symbols: Weighted Overlap and Add (WOLA). The OFDM modulator implemented here follows a CP-OFDM with WOLA approach, and its datapath structure is shown in Figure 1. The main baseband parameters involved in this waveform are the IFFT size $N$, which is equivalent to the number of subcarriers per OFDM symbol, the CP length $L_{CP}$, and the number of time-domain samples used for WOLA– $W$.

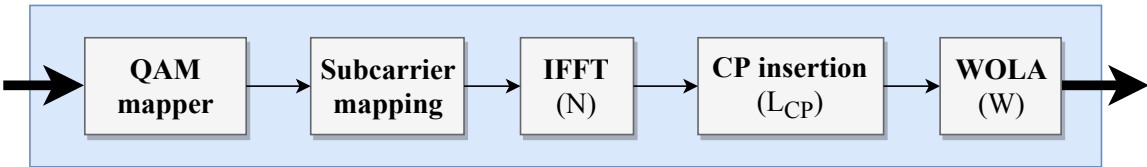

**Figure 1.** Datapath structure for OFDM baseband modulation.

Due to its sinc-pulse shapes transmission, OFDM does not provide genuinely band-limited signals, and the high side lobe power levels can cause unwanted interference with adjacent spectrum bands. FBMC achieves better spectral containment by filtering each subcarrier individually. This eliminates the need for a guard interval like the cyclic prefix and contributes to a higher spectral efficiency. Quite often, the improved spectral efficiency of FBMC systems comes at the cost of relaxed signal orthogonality. In these cases, Offset QAM (OQAM) is employed to ensure real-part orthogonality of FBMC symbols: OQAM-FBMC. In this work, FBMC modulation (Figure 2) follows the approach from [30], where frequency spreading is applied before the IFFT. The frequency spreading operation, which is characterized by the overlapping factor $K$, comprises an up-sampler module and an FIR filter with $2 \times K - 1$ non-zero coefficients. Due to the frequency spreading operation, the waveform synthesis is performed with an IFFT of size $K \times N$, where $N$ is the number of subcarriers. Finally, $K$ IFFT output blocks are overlapped and added to create an FBMC output multi-carrier symbol.

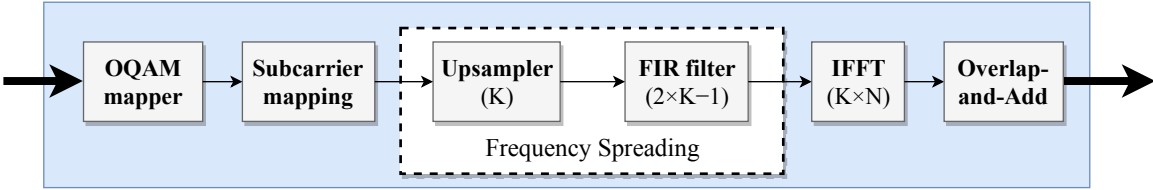

**Figure 2.** Datapath structure for frequency spreading Filter-Bank Multi-Carrier modulation.

UFMC is a waveform with better out-of-band suppression than CP-OFDM and a better multi-antenna compatibility than FBMC [28]. This multi-carrier scheme divides the $N$ available subcarriers that represent the whole frequency band into blocks of subcarriers—Physical Resource Blocks (PRB)—that represent individual sub-bands. Usually, only a part of the PRBs is used for transmission: active PRBs. Then, for each active PRB, IFFT and bandpass $L$-order FIR filtering are

performed. The same filter can be applied to all sub-bands, but its center frequency must be shifted. At the end, the filtered sub-bands are superimposed to form the UFMC symbol to be transmitted. The classic UFMC modulation scheme from [31] considers an $N$-point IFFT and frequency-shifted FIR filters with complex coefficients for each sub-band. To counteract this increased computational complexity, Knopp et al. [29] proposed an algorithm that combines a reduced size $N'$-point IFFT with $\frac{N}{N'}$ upsampling and keeps real-valued coefficient FIR filters by performing frequency shifting after filtering (Figure 3).

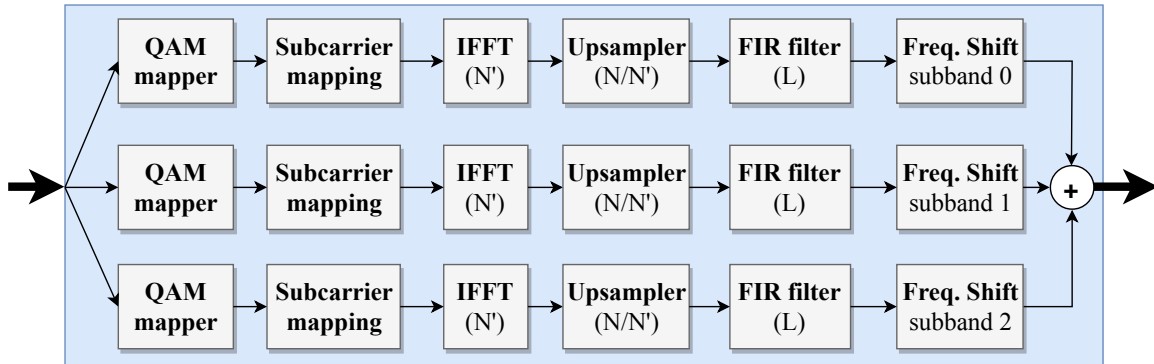

**Figure 3.** Datapath structure for UFMC baseband modulation.

The transition from 4G to 5G will not be as abrupt as in previous generations. Instead, 5G should enable the coexistence and tight interworking between different radio access technologies in order to facilitate the gradual penetration of 5G systems [2]. Thus, the waveform numerologies adopted in our work are based on the 4G LTE standards. In particular, OFDM Modes 1 and 2 correspond to LTE 5 MHz and 10 MHz channelizations, respectively. Like [18], we assume that a primary user communicates using OFDM and that secondary users opportunistically transmit using OFDM, FBMC, or UFMC. Thus, the numerologies for FBMC and UFMC should be compatible with the OFDM numerologies. Table 1 presents the modes of operation and numerologies supported in this work, and Figure 4 depicts the combination of periodograms for Mode 1 OFDM, FBMC, and UFMC baseband signals, in what would be a scenario where these waveforms coexist by sharing a portion of the spectrum band. In all cases, the 16-QAM constellation scheme was used for digital modulation.

Regarding the hardware implementations for the baseband modulator datapaths, we will describe the UFMC modulator design in more detail, and for OFDM and FBMC, we refer to our previous works [22,26]. The implemented modulator datapaths have AXI4-Stream-compatible input/output data interfaces, and all arithmetic operations are done in fixed-point precision, considering real and imaginary parts represented in the Q5.11 format.

The UFMC modulator architecture follows the algorithm description from [29], also illustrated in Figure 3. The first module of a sub-band branch is the QAM mapper. For the 16-QAM case, the module is simply implemented with a 16:1 multiplexer: a four-bit input signal selects the corresponding complex value out of 16 values that compose the constellation. The subcarrier mapping module maps the 12 PRB subcarriers to the central bins of an $N'$-element array and zeroes the remaining $N' - 12$ subcarriers. This module comprises a double buffer of $2 \times N'$ elements and read/write control engines. The double buffer is implemented using dual-port block RAMs embedded in the FPGA logic fabric and allows for simultaneous read and write of consecutive $N'$-element arrays without causing any data conflicts.

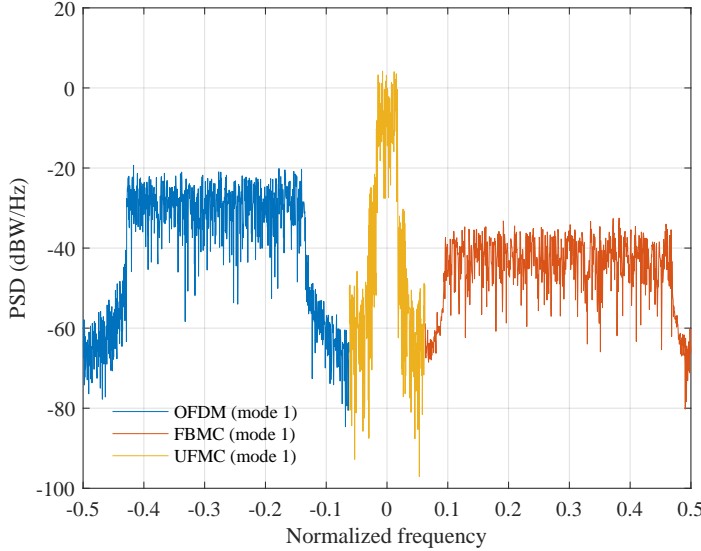

**Figure 4.** Periodograms for OFDM, FBMC, and UFMC baseband signals.

**Table 1.** Waveform numerologies.

**(a)** OFDM

|  | **Mode 1** | **Mode 2** |
|---|---|---|
| # subcarriers, $N$ (IFFT size) | 512 | 1024 |
| CP length, $L_{CP}$ | 40 (1st slot symb.) 36 (other symb.) | 80 (1st slot symb.) 72 (other symb.) |
| WOLA samples, $W$ | 4 | 6 |

**(b)** FBMC

|  | **Mode 1** | **Mode 2** |
|---|---|---|
| # subcarriers, $N$ | 512 | 1024 |
| Overlapping factor, $K$ | 4 | 4 |
| IFFT size, $K.N$ | 2048 | 4096 |

**(c)** UFMC

|  | **Mode 1** | **Mode 2** |
|---|---|---|
| # subcarriers, $N$ | 512 | 1024 |
| # subcarriers per PRB | 12 | 12 |
| # active PRBs | 3 | 3 |
| IFFT size, $N'$ | 64 | 64 |
| Upsampling factor, $\frac{N}{N'}$ | 8 | 16 |
| Filter length, $L$ | 37 | 73 |
| Filter type | Dolph–Chebyshev (60-dB side lobe attenuation) | |

The IFFT computation involves complex arithmetics, and it is replicated for each sub-band processing branch. Consequently, the design choice for the IFFT module should consider a balanced trade-off between performance and resource usage. There are two dominant categories of IFFT/FFT architectures: pipelined and memory-based. In pipelined architectures, the IFFT datapath is tightly synchronized in time and can simultaneously execute transform calculations on the current data frame, load the next input data frame, and unload the results from the previous data frame. This allows

for the continuous flow of data along the datapath at the cost of a higher resource utilization. In turn, memory-based architectures are characterized by an iterative processing nature, and the input data loading and results unloading operations cannot occur simultaneously with transform processing. Compared with pipelined architectures, memory-based architectures consume less circuit area/resources, but provide a lower performance.

Although pipelined-based IFFT modules were adopted in the OFDM and FBMC modulators, we chose a memory-based approach to design the IFFT modules in the UFMC modulator, motivated by three aspects. First, UFMC is a preferred waveform for short-burst transmissions. The iterative nature of memory-based architectures is well adapted to this scenario where the ability for continuous data-stream processing is not a priority requirement. Second, the lower resource utilization of memory-based architectures allows for a more scalable replication of IFFT cores per sub-band branch. Third, memory-based architectures allow for the application of pruning algorithms [32]. Before starting the IFFT processing, the location of the non-zero values within the $N'$-element input array are known. This can be used to prune arithmetic operations between zero values and thus reduce the transform processing time. The IFFT architecture implemented follows a Decimation-In-Frequency (DIF) Radix-2 algorithm, where the processing of an $N'$-point IFFT is divided into $\log_2 N'$ processing stages of $\frac{N'}{2}$ processing steps. The processing steps are executed by a butterfly unit that picks two input values and produces two results: (1) the sum of the two input values; and (2) the difference of the two input values multiplied by a complex twiddle factor.

The memory-based IFFT architecture implemented is depicted in Figure 5, and its main constituent elements are: a control engine, a Radix-2 butterfly unit, two $\frac{N'}{2}$-element memory banks (M0 and M1) and a ROM memory used to control IFFT pruning (pruning ROM). Due to the DIF algorithm employed, IFFT results are not produced in natural order. Therefore, a reordering unit is attached to deliver IFFT output results in natural order to the subsequent datapath modules. The operation of the IFFT module can be divided into two phases: load input/unload results and process transform. During the load input/unload results phase, the control engine issues read/write operations on M0 and M1 to fill the memory banks with the incoming data samples, while forwarding the results from previous transform processing to the reordering unit. The process transform phase corresponds to the execution of the processing steps of each Radix-2 IFFT processing stage. The control unit fetches values from M0 and M1 to the butterfly unit that performs a processing step. Then, the results are stored back in M0 and M1. In this architecture, the control engine uses a binary counter to generate all the signals to control the butterfly unit and memory bank addressing. We adopted the butterfly structure and address generation scheme from [33] and further extended the architecture to support IFFT pruning. The complex multiplier used for twiddle factor multiplications was implemented with three real multipliers, one adder, and two subtractors.

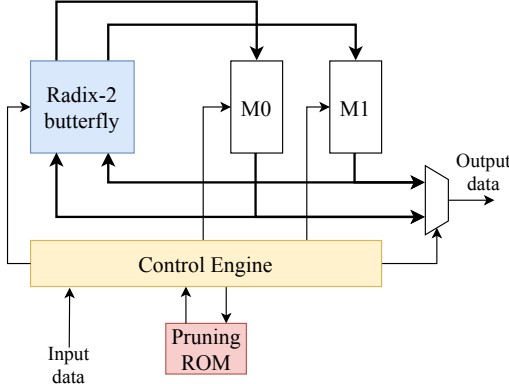

**Figure 5.** Memory-based architecture for the IFFT modules used in the UFMC modulator.

The profile of the IFFT input data array is known in advance: 12 subcarriers are mapped to the central bins of a 64-element array, and the remaining 52 elements are zero. Following the DIF Radix-2 algorithm, it is possible to pre-determine the processing steps that need to be executed and those that can be pruned. The pruning ROM contains information about the number of processing stages where pruning occurs—pruning stages—and for each of these stages, it provides the number of processing steps to be executed, as well as their corresponding control binary counter values to be used by the control engine. For the pruning stages, the control engine fetches the binary counter values from the pruning ROM. When the end of the pruning ROM is reached, the control engine knows that there are no more pruning stages and, thus, internally generates the binary counter values simply by incrementing it from 0 to $\frac{N'}{2} - 1$. In our case, there are two pruning stages comprising 12 and 24 processing steps each. Therefore, the pruning ROM is made of 39 words: one to indicate the amount of pruning stages, two to indicate the amount of processing steps of each pruning stage, and the $12 + 24 = 36$ binary counter values for each processing step. The binary counter word length is eight bits ($\log_2 N' - 1 + \lceil \log_2(\log_2 N') \rceil$, with $N' = 64$), as indicated in [33]. As in [29], the $N'$-point IFFT is followed by upsampling. The upsampler introduces $\frac{N}{N'} - 1$ zeros between consecutive IFFT output samples, and its implementation consists of an FSM alternating between output data and output zero states.

Bandpass FIR filtering for each sub-band is carried by a Dolph–Chebyshev filter with filter length equal to the LTE CP length plus one ($L = L_{CP} + 1$). A FIR filter architecture with a transpose structure was adopted because, unlike the direct FIR model, it does not require an extra input shift register, nor a tree of pipelined adders to achieve high throughput. For the UFMC numerologies from Table 1, the filter lengths are odd, and the coefficients are symmetric with a single center coefficient equal to one. The multiplications by the center coefficient can be ignored, as they do not affect the input value. However, the remaining $L - 1$ coefficients imply non-trivial multiplications. The amount of non-trivial multiplications per FIR filter can be halved ($\frac{L-1}{2}$) by exploiting the coefficient symmetry. As the sub-band signal to be filtered is complex-valued, both real and imaginary parts have to be filtered. Therefore, for each sub-band branch, we have two FIR filters that combine $L - 1$ non-trivial multiplications.

In Xilinx FPGAs, non-trivial multiplications can be efficiently executed by DSP blocks, which are embedded into the logic fabric in a column arrangement. Cost-optimized devices have a smaller amount of DSP blocks, and their utilization should be carefully managed. For instance, the xc7z020 device has 220 DSP blocks, while the total amount of multiplications for FIR filtering in all three UFMC sub-bands ($3 \times (L - 1)$) is 108 for Mode 1 and 216 for Mode 2. The high DSP utilization and its sparse distribution within the logic fabric degrade the scalability of the UFMC modulator. Moreover, it also hampers the place-and-route tasks by EDA tools, affecting overall timing closure.

In these circumstances, we adopted a multiplier-less architecture for FIR filtering where FIR coefficients in the Q1.5 format are represented using the Canonic Signed Digit (CSD) system with minimum non-zero bits. Multipliers are then substituted by shift-and-add graphs. As an example, for a coefficient equal to 0.90625, we have:

$$
\begin{aligned}
0.90625_{10} &= 0.11101_2 \\
&= 1.00\bar{1}01_{2,CSD} \\
&= (1 - 2^{-3} + 2^{-5})_{10}.
\end{aligned}
\tag{1}
$$

Figure 6 illustrates the shift-and-add graph to implement $0.90625 \times x$. This filter design eliminates the use of DSP blocks, but increases slice utilization. Yet, slices are the predominant resource type in the FPGA logic fabric (13,300 slices in the xc7z020 device), which makes our approach viable. After FIR filtering, it is necessary to shift the sub-band signal to the corresponding frequency band. The frequency shift module for each sub-band has a ROM memory to store the complex exponential values and a complex multiplier similar to the one used in the IFFT module. Thus, the overall DSP block

utilization in the UFMC modulator consists of three DSPs in the IFFT and three DSPs in the frequency shift module per sub-band branch. Finally, the filtered sub-band responses are summed to create the aggregate UFMC signal.

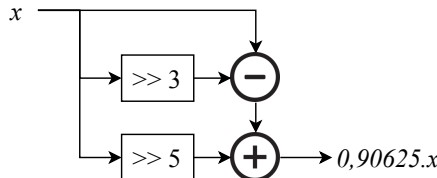

**Figure 6.** Example of a shift-and-add graph to implement a non-trivial multiplication by 0.90625.

### 3.2. Top-Level Architecture

From a top-level perspective, our design (Figure 7) makes use of the hybrid (HW/SW) nature of the Xilinx Zynq architecture that contains two sections: the Processing System (PS) and the Programmable Logic (PL). The PS comprises an ARM Cortex-9 processor and a 512 MB DDR memory controller. The ARM core manages and triggers reconfiguration procedures and sets up data transfers between the DDR memory and the PL. The PL section is then divided into three domains: the baseband processing domain, the DFS domain, and the DPR domain.

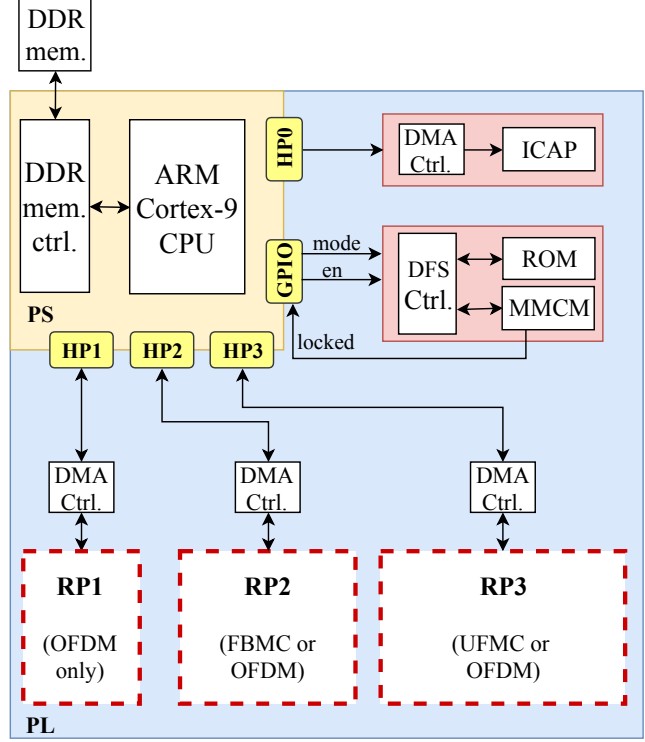

**Figure 7.** Top-level architecture. HPx: High Performance ports, GPIO: General Purpose I/O, RP, Reconfigurable Partition.

The baseband processing domain contains the Reconfigurable Partitions (RPs) that can be dynamically reconfigured to implement different 5G waveform modulators. In this design, three independent RPs are considered: $RP_1$ implements OFDM modulation modes only; $RP_2$ implements FBMC and OFDM modulation modes; and $RP_3$ implements UFMC and OFDM modulation modes. At this stage, an alternative system partitioning strategy could consider an RP for each block in the datapaths or by identifying hardware modules common to all configurations and keeping them in the

system static part (outside the RPs). However, a higher reconfiguration resolution would enlarge the amount of partial bitstreams to store. The implementation of the whole modulators in a single RP also permits the global place-and-route optimization of the processing chain, contributing to an overall smaller reconfigurable area and, consequently, smaller reconfiguration latencies.

When the system is started up, input data files are downloaded from an SD card to the DDR memory. Then, the operation cycle of the baseband processing domain consists of the following steps: (1) fetch input data from, the DDR and feed it to the baseband modulator(s); (2) perform baseband modulation and send the modulated data back to the DDR memory. In a real application, the reconfigurable baseband processor implemented in this work would be integrated with all-digital transceivers such as the ones proposed in [34]. The DMA controller alleviates the PS load related to data transfers to/from the DDR and thus improves baseband processing throughput.

The baseband modulators were designed to run at a clock frequency of 100 MHz. However, through DFS, the clock frequency can be changed at run-time in order to adapt the system to different throughput requirements or power consumption constraints. The DFS implementation follows an approach similar to [23], comprising a Mixed-Mode Clock Manager (MMCM) primitive and a DFS controller engine. The MMCM provides access to the Dynamic Reconfiguration Port (DRP) that allows for writing configuration bits to change MMCM output clocks at run-time. A 100 MHz input reference clock signal provided by the PS (FCLK0) is used by the MMCM to generate an output clock signal for baseband processing purposes. Four MMCM output clock modes are considered: 100 MHz, 66.7 MHz, 33.3 MHz, and 16.7 MHz. The 100 MHz clock frequency was the reference frequency for implementation, while the other values were based on the $2^n$ scaling of the LTE sampling frequency proposed for 5G systems [8]. Applying it to the sampling frequency for 10 MHz LTE channelization (15.36 MHz), we have $2^1 \times 15.36$ MHz $= 30.72$ MHz and $2^2 \times 15.36$ MHz $= 61.44$ MHz. To change the baseband processing clock frequency, the PS defines the MMCM output clock mode through the DFS controller mode port and writes '1' to the en port. After the locked signal becomes active, the baseband modulators are ready for processing.

The DPR implementation provides an infrastructure to access the FPGA configuration memory. In real-time scenarios like wireless communications, it is necessary to reduce the reconfiguration latency because if the system takes too long to reconfigure, quality-of-service is degraded. Moreover, there is also an energy consumption overhead during the reconfiguration, and shorter reconfiguration latencies are crucial to reduce it. The DPR latency depends on the size of the partial bitstreams and on the configuration port bandwidth.

The configuration interface adopted in this work is the ICAP. This high-bandwidth internal interface permits the FPGA to reconfigure itself. Xilinx sets the maximum ICAP bandwidth at 400 MB/s, for a 100 MHz clock frequency and 32-bit data width [35]. Nevertheless, the ICAP can be overclocked to further enhance the reconfiguration throughput [36]. In the present work, the ICAP is overclocked at 200 MHz, using another clock signal (FCLK1) provided by the PS. Like the input data files, the partial bitstreams for all modulator and demodulator configurations are loaded from an SD card to the DDR memory upon system start-up. To take advantage of ICAP overclocking, a dedicated DMA controller is used to accelerate the partial bitstream transfer to the ICAP.

### 3.3. Limitations and Scope

Achieving flexibility in hardware designs for digital signal processing has a cost in terms of power consumption or circuit area [37]. The presented FPGA-based design is not suitable for user terminals in wireless communications, where low-power and low-cost, high-volume production constraints make Application-Specific Integrated Circuits (ASICs) a better solution. Regarding base stations, the combination of General Purpose Processors (GPPs) and Graphics Processing Units (GPUs) yields high flexibility, but requires a high power consumption. Moreover, the performance loss related to GPP-GPU data transfers may be prohibitive in real-time applications [38]. In turn, FPGA's parallelism and programmability are very convenient for flexible and upgradeable base stations with better

processing throughputs than digital signal processors and better power efficiency than GPP- and GPU-based solutions [39].

The high versatility of the proposed architecture is advantageous in different 5G scenarios. For instance, this baseband infrastructure could be used in a multi-service/waveform base station where primary transmissions are 4G OFDM-based (implemented on $RP_1$), while opportunistic secondary transmissions can be based on OFDM, FBMC, or UFMC (implemented on $RP_2$ or $RP_3$). The three independent baseband modulators could also work as part of a multidimensional PHY for CA-based primary communications, where each RP would implement waveform generation for each component carrier to aggregate. Another application scenario to exploit this versatility is centralized baseband processing in C-RANs: the RPs could be dynamically reconfigured to support different radio access technologies and modes operated by RRHs. The multidimensional nature of our baseband infrastructure is a considerable advantage compared to the monolithic approaches from [16,17,19].

Although we have defined only two modes of operation for three waveforms, the functionality can be easily expanded. When implementing the system using the Xilinx Vivado EDA tool, a static-only design checkpoint with locked placement and routing was saved. To upgrade the system with new functionalities, it is only necessary to load new RP configuration designs into the static design check point and generate partial bitstreams for those configurations. The main limitation of the creation of new RP configurations is the RP size and available resources. This design reusability makes the system adaptable and reduces the upgrade design time.

## 4. Results and Discussion

The proposed architecture was evaluated in terms of performance, resource utilization, power consumption, and reconfiguration latency. The results are further discussed within the context of the considered application domain.

The functional correctness of all modulators was verified by checking the simulation results against the values produced by MATLAB scripts [40,41]. The periodograms in Figure 4 were obtained from I/Q samples produced by the implemented modulators operating in Mode 1. Table 2 presents values for the latency—amount of clock cycles to produce the first output sample—for each of the six modulator variants supported. Higher latencies were observed for FBMC, mainly because of the larger IFFT sizes and the time overhead associated with the overlap-and-add operation after the IFFT module. In turn, UFMC showed smaller latencies due to the parallel processing of each sub-band, as well as the smaller IFFT sizes employed. These results are in line with the observations made in [42]: UFMC was more suitable for short packet and low latency transmissions, while FBMC was more efficient for long sequence transmissions. The sub-band parallel processing approach in our design made the UFMC datapath latency independent of the number of sub-bands. Therefore, the latency per sub-band was also 421 clock cycles, which is lower than the 516 clock cycles recorded by Jafri et al. [28]. In steady-state operation and after the initial latency, all the modulators produced one sample per clock cycle. Therefore, their processing throughput was dictated by the clock frequency used for baseband processing.

**Table 2.** Processing latency (in clock cycles).

| Modulator | Mode 1 | | | Mode 2 | | |
|---|---|---|---|---|---|---|
| | OFDM | FBMC | UFMC | OFDM | FBMC | UFMC |
| Latency | 2356 | 7743 | 421 | 5172 | 17,469 | 421 |

Table 3 provides a general overview of the resource utilization for the static and reconfigurable parts of the system. The static part comprised modules for DFS and DPR implementation, DMA access, and communication between the PL and PS sections. This represents less than 32% and 5% of the available slices and block RAMs (BRAMs), respectively. On the other hand, the aggregate amount of

reserved resources for the three RPs accounted for 52.6% of slices, 64.3% of BRAMs, and 72.7% DSP blocks. The exploitation of reconfigurable techniques such as DFS and DPR had an associated resource overhead that is quantified in Table 4. The combined DPR and DFS resource overhead was very small: less than 4% of the slices and around 1% of the BRAMs available in the xc7z020.

**Table 3.** Post-implementation resource utilization for the static and reconfigurable system parts. The figures in parenthesis correspond to the percentage of available FPGA resources.

| Resource | Available (xc7z020) | Static Part | $RP_1$ | $RP_2$ | $RP_3$ | All RPs |
|---|---|---|---|---|---|---|
| Slice | 13,300 | 4210 (31.7%) | 1400 | 2400 | 3200 | 7000 (52.6%) |
| LUT | 53,200 | 10,700 (20.1%) | 5600 | 9600 | 12,800 | 28,000 (52.6%) |
| FF | 106,400 | 13,110 (12.3%) | 11,200 | 19,200 | 25,600 | 56,000 (52.6%) |
| BRAM | 140 | 7.5 (5.4%) | 20 | 40 | 30 | 90 (64.3%) |
| DSP | 220 | 0 | 40 | 80 | 40 | 160 (72.7%) |

**Table 4.** Post-implementation resource overhead for DFS and DPR. The figures in parenthesis correspond to the percentage of available FPGA resources.

| Resource | DFS Overhead | DPR Overhead |
|---|---|---|
| Slice | 24 (0.18%) | 424 (3.19%) |
| LUT | 75 (0.14%) | 938 (1.76%) |
| FF | 79 (0.07%) | 1292 (1.21%) |
| BRAM | 0 | 1.5 (1.07%) |
| DSP | 0 | 0 |

To better understand how the RP resources were actually used, Table 5(a) exhibits the resource utilization per modulator variant. The UFMC modes registered a higher slice and DPS utilization compared to OFDM and FBMC. Most of the slices used in the UFMC modulator implemented the multiplier-less FIR filters. Recalling the DSP block utilization analysis from Section 3.1, the 18 DSP blocks came from the six blocks used in each sub-band branch. The higher BRAM utilization in FBMC modulators was mainly caused by the overlap-and-add of *K* consecutive symbols.

Table 5(b) shows resource utilization for baseband modulators in [19,28]. The operation of the modulators from [19] was defined by setting parameter values, which means that the modules had to be dimensioned for the most resource-demanding mode of operation. For both OFDM and FBMC, the most demanding mode of operation was equivalent to Mode 1 from Table 1. The authors also claimed a maximum clock frequency of 200 MHz for their designs, but only reported post-synthesis results. Additionally, they used LUT RAMs instead of the BRAMs available on the xc7z020 to implement memory elements. Thus, it is hard to do a fair LUT utilization comparison with our work. Still, considering Mode 1, our OFDM modulator used less 30% FFs and 13% DSP blocks, while our FBMC used less 39% FFs and 34% DSP blocks than the corresponding designs from [19].

Regarding UFMC, a direct comparison would reveal that our design used more LUTs, RAMs, and BRAMs than the works from [19,28]. However, a careful analysis of the numerologies is required. The most demanding mode of operation for UFMC in [19] was similar to Mode 1 in our work, except that only one PRB was considered. In our UFMC Mode 1 design, each of the three PRB branches required 2737 LUTs, 2093 FFs, 4 BRAMs, and 6 DSP blocks. Similarly to the OFDM and FBMC cases, it is hard to compare the LUT utilization. Yet, a single PRG branch in our UFMC modulator used 32% FFs and 72% DSP blocks less than the UFMC modulator from [19]. In turn, the UFMC modulator from [28] had a similar numerology to Mode 2 in our work, but considered 25 PRBs processed in a ping-pong buffering/memory fashion. Thus, the architecture consisted of the computational resources required to process one PRB and control structures to allow the continuous processing of several PRBs. The resource consumption per UFMC Mode 2 sub-band in our design was: 3927 LUTs, 3304 FFs, 4

BRAMs, and 6 DSP blocks. As stated previously in this section, the higher LUT and FF utilization per PRB in our design was mainly due the shift-and-add operations on the bandpass FIR filter. On the other hand, the DSP block utilization per PRB was around ten-times lower than in [28].

**Table 5.** Resource utilization for the implemented baseband modulator cores and comparison with other related works.

**(a)** Our work: post-implementation results; device: xc7z020; $f_{clk} = 100\,\text{MHz}$

| Resource | Mode 1 | | | Mode 2 | | |
|---|---|---|---|---|---|---|
| | **OFDM** | **FBMC** | **UFMC** | **OFDM** | **FBMC** | **UFMC** |
| Slice | 994 | 1575 | 2301 | 1139 | 2210 | 3103 |
| LUT | 2940 | 5103 | 8210 | 3395 | 7876 | 11,780 |
| FF | 2107 | 2307 | 6279 | 2170 | 2284 | 9912 |
| BRAM | 7 | 19 | 11.5 | 10.5 | 40 | 11.5 |
| DSP | 14 | 21 | 18 | 14 | 21 | 18 |

**(b)** Related works

| | Nadal et al. [19] | | | Jafri et al. [28] |
|---|---|---|---|---|
| FPGA device | xc7z020 | | | xc7v2000t |
| $f_{clk}$ | Post-Synthesis 200 MHz | | | Post-Implementation 364 MHz |
| Fixed-point precision | 16-bit | | | 16-bit |
| Waveform Numerology | OFDM ∗ | FBMC ∗ | UFMC † | UFMC ‡ |
| LUT | 4511 | 8765 | 5945 | 1133 |
| FF | 3006 | 3788 | 3073 | 910 |
| BRAM | n/a | n/a | n/a | 3 |
| DSP | 16 | 32 | 20 | 64 |

∗ The most resource-demanding mode supported is equivalent to Mode 1 in our work.  † The most resource-demanding mode supported is equivalent to UFMC Mode 1 in our work, except that only 1 PRB is considered. ‡ The supported mode of operation is equivalent to UFMC Mode 2 in our work, except that 25 PRBs are considered. n/a, not available or not applicable.

Besides this comparative analysis, the key observation in our architecture is that, although the three RPs jointly reserved 7000 slices, 90 BRAMs, and 160 DSP blocks, the aggregated amount of resources used by all six baseband modulation configurations was 11,322 slices, 99.5 BRAMs, and 106 DSP blocks. In fact, if we added this amount of resources to the static part resources, we would actually exceed the amount of slices available in the xc7z020 by 17%. As not all baseband modulators operated simultaneously, the hardware virtualization enabled by DPR contributed to the increased functional density and resource efficiency of the proposed architecture. Assuming that all three RPs were in use, the currently supported modes of operation allowed for 32 possible combinations to arrange the baseband modulators: 2 $RP_1$ modes $\times$ 4 $RP_2$ modes $\times$ 4 $RP_3$ modes. However, this versatility can be augmented by adding new RP configurations for waveforms and mode operations needed in the future, extending the system duty lifetime. This is a clear advantage over the design from [19]: although supporting the same waveforms, it is not easily upgradeable due to its static nature. Additionally, the modulator redesign in [19] to support new modes of operation was strictly confined to the available FPGA resources, as no hardware virtualization techniques were employed.

Another relevant aspect is cost-effectiveness. Using an FPGA device with a larger chip area, it would be possible to implement a wide range of baseband modulator modes with relaxed resource availability constraints. However, a larger chip area is more expensive and typically consumes more power. In the proposed architecture, the use of more hardware resources than those available on the

FPGA device is possible. This enables the system implementation on a small form, cost-optimized device with immediate benefits regarding cost and power consumption compared to larger devices [5].

The clock frequency adaptation through DFS will affect dynamic power consumption. Power estimates for baseband processing were obtained using the Xilinx Vivado 2015.2 Power Analysis tool. To increase the estimates' confidence level, post-place-and-route power analysis with node activity derived from simulation files was performed. For each modulator variant, dynamic power was estimated considering the clock frequency modes defined for DFS (Table 6). Modulator variants with higher resource utilization are likely to induce more node activity and consequently have a higher dynamic power consumption. This justifies the higher power consumption of UFMC modes compared to FBMC and OFDM. The different node activity for each modulator is also the reason why power does not downscale linearly with frequency in every case.

**Table 6.** Dynamic power consumption estimates for the six implemented baseband modulator cores (in mW). Device: xc7z020; analysis tool: Vivado 2015.2; post-implementation power analysis with a high confidence level; node activity derived from post-implementation simulation.

| $f_{clk}$ | Mode 1 | | | Mode 2 | | |
|---|---|---|---|---|---|---|
| | **OFDM** | **FBMC** | **UFMC** | **OFDM** | **FBMC** | **UFMC** |
| 100 MHz | 113 | 148 | 180 | 123 | 161 | 233 |
| 66.7 MHz | 74 | 84 | 119 | 78 | 79 | 155 |
| 33.3 MHz | 34 | 25 | 60 | 33 | 28 | 77 |
| 16.7 MHz | 14 | 8 | 30 | 10 | 10 | 39 |

Without DFS support, the clock frequency for baseband processing would be fixed to cover the most demanding throughput scenario, even when a lower throughput would be enough. In our design and considering low-throughput scenarios satisfied by a 16.7 MHz clock frequency, DFS allows for power reductions between 99 mW (OFDM Mode 1) and 194 mW (UFMC Mode 2) compared with the operation at 100 MHz. These values represent relative reductions of 88% and 83%, respectively. Comparing operation at 16.7 MHz and 33.3 MHz, we observe power differences from 20 mW (59% reduction in OFDM Mode 1) to 38 mW (49% reduction in UFMC Mode 2). Thus, in our architecture, DFS not only allows for run-time processing throughput adaptation, but also improves the baseband processing power efficiency.

Changing the functionality of an RP or the clock frequency used for baseband processing does not occur instantaneously. In real-time systems, it is important to measure and evaluate the impact of reconfiguration latency. With a 100 MHz input reference input clock, a DFS procedure takes on average 47 μs to modify and lock the baseband processing clock frequency. DPR latency was measured for each RP, and the worst-case scenarios are displayed in Table 7. To reduce partial bitstream sizes, they were compressed. As expected, the RP with larger area—RP$_3$—had a longer reconfiguration latency associated. Still, the reconfiguration of an RP always took less than a millisecond. In all experiments, the reconfiguration speed was at least 790 MB/s, which is around 99% of the maximum theoretical throughput for the ICAP, considering 32-bit data transfers at 200 MHz.

**Table 7.** DPR latency (worst-case scenarios).

| | RP$_1$ | RP$_2$ | RP$_3$ |
|---|---|---|---|
| DPR latency | 400 μs | 677 μs | 767 μs |
| Partial bitstream size | 309 kB | 526 kB | 596 kB |

Reconfiguring the communication mode of operation would typically be carried out at the control plane. An ITU-R report about technical performance specifications for 5G radio interfaces recommends control plane latencies below 10 ms [43]. Therefore, the latencies observed in our architecture are

acceptable. Nonetheless, in critical, high-priority scenarios, our architecture allows a make-before-break approach to eliminate DPR latency: use a spare RP to load the new transmission mode before the old one is terminated. This is illustrated in Figure 8: OFDM Mode 1 communication carried out in RP1 has to be adapted to OFDM Mode 2; the idle RP2 is reconfigured to OFDM Mode 2 before communication using RP1 terminates. Then, the communication handover from RP1 to RP2 is not affected by the DPR latency.

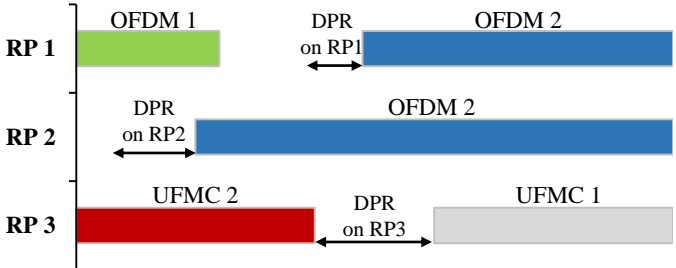

**Figure 8.** Example of the make-before-break approach to mitigate DPR latency.

## 5. Conclusions

This paper proposes a reconfigurable FPGA-oriented baseband modulator architecture suitable for 4G/5G scenarios, such as non-contiguous carrier aggregation, parallel multi-access processing in C-RANs, and multiple waveform coexistence. The design was implemented on a Zynq xc7z020 device and supports three waveforms: OFDM, FBMC, and UFMC. It considers three independent, dynamically reconfigurable baseband processing blocks, whose mode of operation and clock frequency can be adapted at run-time through DPR and DFS. The clock frequency adaptation at run-time enables throughput adjustment according to communication demands and improves overall power efficiency (power reductions from 49%–88%) at the cost of a negligible resource and latency overhead (47 μs). The aggregate sum of FPGA logic slices used for all modulator configurations plus the slices used for the system static part exceed the available slices on the xc7z020 by 17%. This enhanced resource efficiency is due to DPR hardware virtualization and makes it possible to implement the proposed architecture on a small-form cost-optimized device. The versatility of the current architecture is highlighted by the 32 possible modulation arrangements, when all three independent baseband processing blocks are simultaneously active. With an acceptable worst-case reconfiguration latency of 767 μs, the application of DPR also makes the architecture forward-compatible with future modes of operation.

**Author Contributions:** Conceptualization, M.L.F. and J.C.F.; methodology, M.L.F. and J.C.F.; validation, M.L.F. and J.C.F.; investigation, M.L.F. and J.C.F.; writing, original draft, M.L.F.; writing, review and editing, J.C.F; supervision, J.C.F.

**Funding:** This work is financed by the ERDF (European Regional Development Fund) through the Operational Programme for Competitiveness and Internationalization (COMPETE) 2020 Programme within Project POCI-01-0145-FEDER-006961, by National Funds through a Ph.D. Grant (PD/BD/105860/2014) from the FCT (Fundação para a Ciência e a Tecnologia) (Portuguese Foundation for Science and Technology).

**Conflicts of Interest:** The authors declare no conflict of interest. The funders had no role in the design of the study; in the collection, analyses, or interpretation of data; in the writing of the manuscript; nor in the decision to publish the results.

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
