# Peer review of "An FPGA-Oriented Baseband Modulator Architecture for 4G/5G Communication Scenarios"

_electronics, doi:10.3390/electronics8010002_

Round 1

Reviewer 1 Report

The article proposes and evaluates an FPGA-oriented baseband modulator architecture for 4G/5G communication scenarios. The paper is overall well written. The reviewer only has some suggestions below:

The proposed architecture is based on FPGA, which is usually of higher cost. The authors can briefly discuss and compare this aspect compared with the other existing architecture.

In Fig. 2, the notations 2.K-1 and K.N seem to be inaccurate.

On page 5 line 194, what does the notation mean in "- N -" and "- L_{CP} -"?

On page 11 line 376, should "MiB/s" be "MB/s"?

On page 9, eq. (1) needs a period.

Author Response

Detailed Answers to reviews:

1) The proposed architecture is based on FPGA, which is usually of higher cost. The authors can briefly discuss and compare this aspect compared with the other existing architecture.

This aspect is now briefly addressed in the newly created sub-section 3.3 (“Limitations and Scope”).

2) In Fig. 2, the notations 2.K-1 and K.N seem to be inaccurate.

Figure 2 was corrected to use the same notation as in the body text (“2×K1” and “K×N”).

3) On page 5 line 194, what does the notation mean in "- N -" and "- L_{CP} -"? (it is written there).

These notations are defined in lines 193-195: “The main baseband parameters involved in this waveform are the IFFT size - N -, that is equivalent to the number of subcarriers per OFDM symbol, the CP length – L_{CP} – and the number of time-domain samples used for WOLA - W.”

4) On page 11 line 376, should "MiB/s" be "MB/s"?

It should indeed be “MB/s”. This was corrected in the reviewed version. We also present our measured reconfiguration throughput in MB/s.

5) On page 9, eq. (1) needs a period.

Corrected.

Reviewer 2 Report

The paper definitely has a merit. It is clearly written and well organized. I appreciate the solid overview of the literature, as well as conducted experiments regarding evaluation of the presented technique.

I have just two issues to be considered (the second one applies for future works):

Does the presented solution have any limitations? Maybe it would be good to add "limitations and scope" section.

I strongly recommend hardware verification of the presented method, especially regarding DPR. Very often theoretical assumptions do not coincident with hardware results. Therefore, please consider hardware measurements for example with the use of oscilloscope to verify the state of signals (for example during DPR). This comment is rather oriented for future works, since in my opinion the current version of the paper is good enough, as it is.

Author Response

Detailed Answers to reviews:

1) Does the presented solution have any limitations? Maybe it would be good to add "limitations and scope" section.

These aspects are now briefly addressed in the new sub-section 3.3 (“Limitations and Scope”), which reuses some material that was already in the paper.

2) I strongly recommend hardware verification of the presented method, especially regarding DPR. Very often theoretical assumptions do not coincident with hardware results. Therefore, please consider hardware measurements for example with the use of oscilloscope to verify the state of signals (for example during DPR). This comment is rather oriented for future works, since in my opinion the current version of the paper is good enough, as it is.

We will consider your recommendation in future works. Thank you!
